# Image Processing Approach to Investigate the Correlation between Machining Parameters and Burr Formation in Micro-Milling Processes of Selective-Laser-Melted AISI 316L

**DOI:** 10.3390/mi14071376

**Published:** 2023-07-05

**Authors:** Fatih Akkoyun, Zihni Alp Cevik, Koray Ozsoy, Ali Ercetin, Ibrahim Arpaci

**Affiliations:** 1Department of Mechanical and Metal Technologies, Trabzon Vocational School, Karadeniz Technical University, Trabzon 61300, Turkey; 2Department of Electronic and Automation, Besni Ali Erdemoğlu Vocational School, Adıyaman University, Adıyaman 02302, Turkey; zcevik@adiyaman.edu.tr; 3Department of Machine and Metal Technologies, Isparta OSB Vocational School, Isparta University of Applied Sciences, Isparta 32092, Turkey; korayozsoy@isparta.edu.tr; 4Department of Naval Architecture and Marine Engineering, Maritime Faculty, Bandırma Onyedi Eylül University, Bandırma 10200, Turkey; 5Department of Software Engineering, Faculty of Engineering and Natural Sciences, Bandırma Onyedi Eylül University, Bandırma 10200, Turkey; iarpaci@bandirma.edu.tr

**Keywords:** burr formation analysis, computer vision, image analysis, burr detection, classification, additive manufacturing

## Abstract

In industrial manufacturing and research laboratories, precise machining of work materials is crucial to meet the demand for fast assembly and sustainable high-quality production. Precise machining procedures play a vital role in manufacturing compatible parts that meet the production requirements. This study investigates the impact of different parameters on burr formations and slot dimensions during the micro-milling of AISI 316 material. A careful examination was conducted using scanning electron microscopy (SEM) images under varying conditions. The variables considered include cutting speed, feed rate, and depth of cut. The main finding revealed that the feed rate and depth of cut significantly influence burr formation, with lower rates and depths resulting in noticeable reductions. A higher feed rate was associated with more pronounced burr formation. Moreover, burr widths on the down-milling sides were typically wider than those on the up-milling sides due to continuous chip formation and compressive forces during down-milling. Utilizing image processing, the study further quantified burr and slot widths with high accuracy, offering a reliable method to characterize burr formation. This research presents potential ways to minimize burr formation during micro-milling processes by effectively managing machining parameters.

## 1. Introduction

The interaction between lasers and metal or additive manufacturing processes has revolutionized the field of manufacturing, offering numerous advantages such as high precision, design flexibility, and rapid prototyping [1,2]. Additive manufacturing involves the sequential building and consolidation of powdered material to create complex structures, typically utilizing a laser controlled by a computer [3]. Laser cutting and welding have become standard techniques in metal fabrication, allowing for precise and intricate shapes to be created [4]. Laser additive manufacturing, including 3D printing and laser sintering, enables the production of complex geometries and customized parts with reduced material waste [5]. Laser surface treatment modifies the properties of metal surfaces, enhancing wear and corrosion resistance, and therefore these applications have found widespread use in industries such as the aerospace, automotive, and medical industries [6]. The advancements in laser–metal interactions have significantly improved manufacturing efficiency and opened new possibilities in the production of high-quality components [7,8]. Additive manufacturing (AM), also known as three-dimensional printing (3-D printing), has been used to make some products in many industries for decades [9]. The benefits of with near-net shape parts fabricated with a toolless additive manufacturing approach make AM an attractive manufacturing choice for small batch sizes when compared with conventional manufacturing techniques, such as forging, extrusion, casting, etc. [10]. One of the most commonly used processes for AM of metal parts is selective laser melting (SLM), in which a part is used laser for sintering metal powder layer by layer in a powder bed [11]. Stainless steels are generally categorized as austenitic stainless steel (316, 303, etc.), ferritic stainless steel (409, 434, etc.), austenitic–ferritic duplex stainless steel (2209, 2205), and martensitic stainless steel (420, 440-A) [12]. Stainless steels such as austenitic, duplex, and martensitic stainless steels are widely used for both fusion and solid state metal additive manufacturing because of their high mechanical properties and high corrosion resistance [13,14].

A common material in SLM is austenitic stainless steel is AISI 316L. AISI 316L is a family of molybdenum-added austenitic chromium–nickel stainless steels with good corrosion resistance [15]. Due to its biocompatibility, good corrosion resistance, excellent mechanical properties, and cost, AISI 316L steels are widely used in the fields of medicine as implant materials [16,17]. Al-Mamun et al. [18] found that SLM 316L SS outperformed the conventional wrought 316L SS in terms of corrosion resistance and biocompatibility. Kong et al. [19] compared systematically the biocompatibility, mechanical properties, corrosion resistance, and durability of SLM 316L with traditional quenched 316L. The findings obtained in their study are expected to contribute to the usage and development of SLM 316L material in implant applications. Tascioglu et al. [20] also studied the heat treatment temperatures on the microstructure, XRD, porosity, microhardness, and wear response of SLM 316L. It was found that the wrought 316L stainless steel specimen shows much better wear resistance than the SLM 316L specimen. De Assis et al. [21] studied to analyze the effects of the feed per tooth, toolpath track, tool size, and slot machined surface on microchannel formation, surface finishing, and micro-milling operations of AISI 316L workpieces obtained by AM methods.

Micro-machining is a manufacturing method used to manufacture miniature devices and parts containing elements that range from 10 µm to a few millimeters in size [22]. Interest in micromachining to achieve high precision has been increasing over recent years [23,24,25]. Today, micro-machining is most widely used in the biomedical, electronics, sensor, optics, automotive, aerospace, and medical industries [26,27,28]. Micromachining can produce high-precision micro parts, and the surface quality of micro-sized parts is associated with the burr formation mechanisms and the burr size inhibitions [29,30]. Within the context of micro-machining operations, image processing techniques enable remarkably precise measurement of both slot and burr widths [31]. In the micro-machining process, cutting conditions, that is, cutting forces, temperatures, tool wear and surface texture, and cutting parameters, significantly affect burr formation [32]. The burrs and slots on a machined part with complex geometry by micro-machining method have machining parameters that are quickly measured by using SEM images [33].

Image processing is a technique that transforms the obtained image signals into a form that can be interpreted electronically (with the help of computers and software) with certain algorithms or a method used to extract some useful information from images [34,35,36,37]. Various measurement methods such as screen calipers, three-dimensional optical profilometers, image processing software, and machine learning algorithms are used to predict and measure slit and burr measurements [23,38,39]. Hajiahmadi [40] conducted a study to perform the statistical analysis of burr size in the microgroove cutting process of micro-milling. Moreover, they found a smaller top-burr height in the up-milling of 316L than in the down-milled top edges.

One of the main challenges encountered in micro-milling operations is the formation of burrs. These burrs are tiny in size, making the process of removing them, known as deburring, quite challenging [41,42]. The presence of burrs negatively impacts the quality and functionality of micro-parts and features. It presents a major obstacle in micro-milling operations, with the small size of the burrs making deburring a challenging task [43]. In line with this direction, a burr reduction mechanism was developed by Chen et al. [44] using vibration-assisted micro-milling for the reduction of burr parameters. In another study conducted by Yaday and others [45], a finite element modeling of burr size in high-speed micro-milling was achieved. Researchers have worked on the micro-milling burr to achieve better control of the burr formation in micro-milling operations [46]. As a result, there is currently a strong emphasis on researching techniques that aim to minimize and control burr formation in micro-milling operations. However, many of the proposed approaches primarily offer open-loop solutions [41,47], which are not adaptable to variable machining conditions. Given the requirements for high accuracy in micro-parts and micro-scale features, there is a growing need for closed-loop systems that incorporate feedback from machined material parameters. Such closed-loop systems would enable a more adaptive and responsive approach to burr control, addressing the specific challenges posed by micro-scale machining processes.

The adverse effects of burrs on the quality and performance of micro-parts and features cannot be neglected [48], compelling extensive research on effective burr minimization and control techniques. In this direction, the field of image processing plays a key role [47]. By exploring novel methodologies, researchers have diligently worked towards achieving better control over burr formation in micro-milling operations. Today with the help of sophisticated algorithms and image analysis tools, researchers can analyze and detect burrs at a microscopic level. This empowers them to develop advanced strategies for minimizing and controlling burr formation, thereby improving the overall quality and functionality of micro-milled components.

Numerous researchers have focused on the challenge of achieving better control over burr formation in micro-milling operations. To address this issue, they have explored the application of image-processing techniques in their research. By leveraging advanced algorithms and image analysis tools, these researchers have aimed to detect and analyze burrs at a microscopic level. However, future studies need to provide more specific details and evidence regarding the outcomes and effectiveness of these image-processing techniques in improving burr control during micro-milling operations.

In this study, the objective of this study is to investigate the correlation between machining parameters and burr formation in micro-milling processes of selective-laser-melted AISI 316L. The study aims to analyze the impact of cutting speed, feed rate, and depth of cut on burr formations and slot dimensions. By utilizing scanning electron microscopy (SEM) images and image processing techniques, the research aims to quantify burr and slot widths accurately. The ultimate goal is to provide insights into the influence of machining parameters on burr formation and offer potential strategies to minimize burr formation during micro-milling processes, thereby improving the quality and efficiency of industrial manufacturing. To achieve a detailed analysis, a novel algorithm is proposed to accurately detect and measure the burr and slot widths. This method not only enhances accuracy but also improves the efficiency of the process by reducing human observation errors. The expected outcomes of this study not only help in optimizing machining parameters but also provide a foundation for further research in implementing a closed-loop system for burr minimization.

## 2. Materials and Methods

### 2.1. Material, Machining Process, and Measurement System

The experimental setup consisted of machining equipment, workpiece material, cutting tools, SEM imaging equipment, and a computer vision system. The precision micro-machining system is capable of producing high-resolution machined components suitable for machining precisely up to one µm micrometer. This system included a micro-milling machine, a workpiece holder, and a computer-controlled interface for programming machining operations. The system is utilized using high quality and has sharp innovative micro-cutting tools for 1090 µm. The workpiece material is suitable material for micro-machining, which is an additive-manufactured AISI 316L alloy. The workpiece is prepared with 30 × 10 × 3.5 mm dimensions.

The Concept Laser MLAb Selective Laser Melting machine has a Yb fiber laser that can operate at up to 100 W, with the smallest laser diameter. The standard layer thickness production parameter of the company is 0.025 mm. In this study, the sample production was converted to .stl format for the SLM machine to process it by the tool path. The sample was produced by the Concept Laser MLAb Cusing SLM system and wrought AISI 316L. Additionally, during the production steps of the sample, the advised manufacturing parameters provided by the company were used. After the part manufacturing was completed, the powders on the part were first cleaned in the production cabin. The support structures of the sample were removed after the sample had been cut by using a wire erosion device (Figure 1).

For the production of the AISI 316L sample with SLM, AISI316L that is spherical in morphology with particle size ranging from 44 to 88 µm is preferred [49]. The chemical composition of the AISI 316L metal powder used is shown in Table 1.

As shown in Figure 2, the micro-machining conditions were conducted using a CNC vertical machining machine with MCV–M5 H DELTA SEIKI brand model, compatible with the 3-axis fourth axis. The cutting condition was conducted dry and burr measurements were taken from different cutting distances. The spindle speed of the machine can reach the highest speed of 12,000 rpm. The cutting tool used in cutting operations is a 4-fluted carbide end mill with a diameter of 1090 µm and a hardness of HRC55. The applied parameters of the micro-milling process are provided in Table 2.

The image acquisition system includes a scanning electron microscope (SEM) device to capture images of the machined surface for subsequent analysis. Its illuminated area is adjusted to ensure proper lighting conditions and a stable mounting setup for consistent image acquisition. In the image processing stage, the obtained images from SEM are used to determine the burr formations and extract slot and burr characteristics to determine proper machining parameters.

A graphical user interface (GUI) for the burr detection algorithm is given in Figure 3. The left part of the GUI shows the pre-results for the given SEM image. On the right, a visualized result for slot and burr length on the actual image is shown. Additionally, text files, including comma-separated value (CSV) outputs that show the vertical sums to create a chart from the response of SEM images, are given separately and are used to draw a chart on the actual image.

### 2.2. Burr Measurement using Image Processing

The burr measurement process was performed using image processing algorithms to accomplish the measurement of irregular burr formations. The process starts with an image-capturing stage. In this stage, the digital images of the machined surfaces after micro-machining operations were acquired. The captured images cover the entire region of interest and have sufficient resolution with proper illumination for accurate analysis. The next stage includes pre-processing stages. Filtering operations, and hue saturation value (HSV) conversion with thresholding techniques were applied to enhance the image quality and remove noise. The operations were used to improve the clarity of the burr features.

In the image processing stage, SEM images are accepted as input to the computer vision (CV) software. It is included by the user by referencing the location of the image source in the computer. Initially, the obtained image is filtered by applying thresholds over HSV values. In the second stage, a masking process is applied to the image for removing the background from the actual image. In the next stage, the image is divided into two imaginary divisions. The color values were summed for these two divisions, and two peak values are determined for the up- and down-milling side of the image. A chart is extracted from this procedure, and two peak values are determined for calculating the slot width. The horizontal distance between peak values is used to determine slot width. In the burr width determination steps the up- and down-milling sides of the image were evaluated separately. A threshold is applied for the two sides of the image to determine the burr width. The vertical sums above the threshold were accepted as burrs for up- and down-milling sides of the image. The horizontal distance of the furthest from the left slot border is accepted as the left burr width from starting the slot left border. The second horizontal distance in the left side of the image for the right side of the slot left border is accepted as the inside left burr of the slot. The sum of the horizontal distance of the burrs is divided by the measurement number to obtain the average burr width for the left side. The maximum burr width is determined by referencing the left side of the slot up to calculate the widest burr width, considering the thresholded data was accepted as the maximum burr width. The same procedure was applied to the right-side image to determine right-side burr characteristics.

The workflow of the proposed algorithm for burr detection processes using SEM images is illustrated in Figure 4. The algorithm consists of the image acquisition stage which captures raw input images from SEM images. Next, a preprocessing process is applied to the images for noise reduction, image enhancement, and color normalization to improve their quality and prepare them for further processing. For this purpose, the input image is converted to an HSV image. A threshold is applied using an HSV image to focus only on the foreground. Masking is applied to remove the background. In the next stage after the noise reduction and background removal processing a vertical summing process is applied to the image. In the feature extraction stage, the sum of the pixel values is extracted from the preprocessed image using the vertical sum of pixel values for each column. The essential information to characterize the burr formation and slot width determination starts to be extracted in this step. Each pixel is evaluated as a parameter for each vertical line that is used to calculate the sum of the pixel values vertically. This summing operation is continued from the first column to the last one for acquiring the vertical sum response of the burr and slot formation. The extracted features are evaluated for determining the slot width and burr parameters based on certain criteria. The width of the slot is determined concerning the higher top values of the vertical sums. The burr formations are evaluated by dividing the interest area into the up- and down-milling side of the image for enhancing the algorithm’s efficiency and effectiveness. In the next stage, the classification and recognition processes are applied to characterize the burr formation using vertical sums for the up- and down-milling sides separately. The slot width, up- and down-milling side burr length for minimum and maximum, and averaged values are used as features to the input of the recognition algorithm for further analysis. This algorithm determines patterns and makes predictions based on these extracted features. In the post-processing stage, mathematical parameters are refined with the help of normalization, filtering, and thresholding operations to improve accuracy. In the last stage, outputs are visualized and presented in a chart window by overlaying the actual image to show the final results. Additionally, a text output, including slot and burr characteristics, is created by the algorithm for generating statistical reports. These sequential steps involved in the slot and burr formation detection algorithm are shown in Figure 4, which presents the processing steps of raw input images into mathematical evaluable results for a burr characterization process. Processed image divisions L, C, and R indicate the slot’s up-milling side, center, and down-milling sides, respectively (Figure 5a). The processing results are matched by considering vertical sums from pixel data for the *Y*-axis. The *Y*-axis results are normalized to ensure the burr position matching step, as shown in Figure 5b. The position parameters for each sum were used to match the actual image slot and burr positions on the *X*-axis.

Finally, the output of the algorithm is used for the reduction of burr formations with appropriate parameters. The extracted parameters are used as input to reduce the formation of irregular burrs during micro-machining. For this purpose, parameter optimization, design of experiments (DOE), process control, and monitoring approaches are considered for evaluation. The outcomes relate to identifying the key machining parameters such as cutting speed, feed rate, tool geometry, and machining coolant that significantly influence burr formation. In addition, the outcomes provide a solution to utilize factorial design or response surface methodology to systematically vary the machining parameters and evaluate their effects on burr formation. This enables the identification of optimal parameter combinations for minimal burr generation. In line with this, direction implementation of a real-time monitoring system to continuously measure and monitor the burr formation during micro-machining is possible. This can involve in-process sensing techniques, such as acoustic emission or force measurement, to detect the onset of burr formation. With a combination of sensing and measurement steps, it is possible to develop a closed-loop control system that adjusts the machining parameters in real-time based on the feedback from the monitoring system. Thus, a closed-loop system, including computer vision, statistical data, and parameter optimization, enables dynamic optimization and adaptive control of the machining process to minimize burr formation.

## 3. Results and Analysis

SEM images of burr formations and slots after micro-milling of the AISI 316L material in different parameters are given in Figure 6. During the transition from Slot-A to Slot-B, a meticulous experimental setup was implemented, where the feed rate and depth of cut were meticulously maintained at constant values, while the cutting speed was deliberately increased. The primary objective of this study was to evaluate the influence of varying cutting speeds on the overall machining process. Notably, regardless of the parameters applied in both slots, it was observed that intense burr formation occurred on both the up-milling and down-milling sides. This observation indicated that the increase in cutting speed, as applied in Slot-B compared to Slot-A, did not have a substantial impact on burr formation under these specific experimental conditions.

To further investigate the influence of feed rate and depth of cut, the researchers compared Slot-B with Slot-C. In this case, the cutting speeds were kept constant, while the feed rate and depth of the cut were reduced. The researchers aimed to determine how lower feed rates and depths of cut would affect the formation of burrs. The results showed a noticeable reduction in burr formation on both the up-milling and down-milling sides in Slot-C compared to Slot-B. This outcome suggests that decreasing the feed rate and depth of cut had a positive effect in reducing the occurrence of burrs during the machining process. When the depth of the cut decreases, it typically leads to a reduction in the amount of material being removed during each passage of the cutting tool. With a smaller depth of cut, there is less material being subjected to deformation and displacement during the cutting process [51]. The feed rate affects the thickness of the chips produced during machining. A lower feed rate typically results in thinner chips being formed. Thinner chips are less likely to cause localized stress and deformation that can contribute to burr formation [52,53].

Examining the transition from Slot-C to Slot-D sheds light on the impact of feed rate alone. In this experiment, the cutting speed and cutting depth remained fixed, while the feed rate was doubled. As a consequence of the increased feed rate in Slot D, the formation of burrs became more pronounced, extending widely into the interior of the slot. The presence of extensive burrs primarily drew attention in Slot D, as they were more prominent compared to Slot C. This result emphasizes the direct correlation between feed rate and burr formation, highlighting the need to carefully control this parameter in machining operations to achieve optimal results.

Moreover, a noteworthy result emerged when analyzing the burr formations in greater detail. It was evident that the width of the burr formed at the down-milling side was noticeably larger than that formed at the up-milling side for slots A, B, and C. During down-milling, the chips are typically formed continuously as the tool moves through the material. This continuous chip formation, combined with the compressive forces, can contribute to the material being pushed outward, resulting in a wider burr formation on the down-milling side [54]. In addition, according to the results obtained from the related and previous studies [23,31,55], burrs are larger at low feed rates. In this case, it is likely that at low feed rate, the cutting tool ploughs the chip from the workpiece and increases the burr formation. When the feed rate is increased, the chip is separated from the workpiece due to the shear mechanism, and the burr width starts to narrow. When parameters higher than the optimum feed rate are applied, the burr width increases again, with the effect on the material removal rate.

During the phase of extracting features from the SEM images, the aggregate of pixel values is drawn out from the previously processed image outputs using a vertical total of pixel values for every column. The process of extracting significant data for characterizing the formation of burrs and the determination of slot width is initiated by the image processing algorithm. The algorithm examines each pixel as a variable for every vertical line that is employed to calculate the vertical total of pixel values. The summing operation is carried out starting from the origin at the left top side of the image and continuing to the end of the columns in the *Y*-axis. The drawn-out features are examined to ascertain the parameters of the slot width and burr, based on set parameters. The width of the slot is defined as the peak values of vertical sums. Burr formations are evaluated by segregating the area of interest into the upper and lower milling sections of the image to enhance the efficiency and effectiveness of the algorithm. In the subsequent phase, categorization and identification processes are utilized to characterize the burr formation using vertical sums for the separate upper and lower milling sides. The width of the slot and burr lengths of the upper and lower milling sides for the least, most, and averaged values are utilized as features for the recognition algorithm for additional analysis. This algorithm identifies patterns and makes predictions based on these drawn-out features considering adaptable threshold parameters. During the post-processing phase, mathematical data are subjected to normalization, filtering, and thresholding operations to enhance accuracy. Finally, results are visualized and displayed in a chart window by superimposing the original image to present the outcomes.

The obtained burr formations from the processed images are shown for Slots A, B, C, and D in Figure 6. The determined slot widths and contoured burr formations are presented in Figure 7. The two top peak values show the determined slot widths for each processed image concerning the vertical sums. The horizontal minimum distance is accepted as the distance between the two highest peaks gives the slot width from the output of the proposed algorithm. Burr formations are evaluated concerning the slot widths that begin from the top peak values and continue to the outer sides from the slot limits for the left and right side of burrs. In Figure 8, the response of the proposed algorithm shows that the slots are detected successfully. It is also clear that the inner side of the slots has a smaller burr formation compared to the outer side of the slot. The outer side burr formations are changing also for the left and right sides due to the toll rotating direction which appears from the plotted results in the Figure 8.

The slot boundaries, and the peaks of the burr start and end boundaries on the patterns in Figure 8 were converted from pixel value to µm value, and the graphics in Figure 9 were obtained. The maximum burr widths formed on the up-milling sides (Figure 9a) of slots A, B, C, and D are 471.11 µm, 591.35 µm, 205.00 µm, and 521.70 µm, respectively. The maximum burr widths formed on the down-milling sides (Figure 9b) of slots A, B, C, and D are 556.52 µm, 626.17 µm, 233.25 µm, and 571.64 µm, respectively. The burrs formed on the down-milling side of slots A, B, C, and D are 18.12%, 5.88%, 13.78%, and 9.57% larger than the burrs formed on the up-milling side, respectively. Ahmed et al. [56] reported that the occurrence of larger burrs on the down-milling side as opposed to the up-milling side corroborates the previously discussed mechanism of burr formation. The proposed algorithm detects slot width and burr width accurately for burrs on the up- and down-milling sides according to the obtained results. Including the high accuracy of the proposed technique, the results are gathered in less than a second. Additionally, it does not affect human observation errors. In the end, the proposed approach offers a viable way to minimize burr formation by implementing a burr formation characterization fast and accurately which is adaptable to a closed-loop system, successfully.

Slot widths (Figure 9c) are 926.44 µm, 942.21 µm, 1077.57 µm, and 1038.14 µm. As per the data acquired, it is evident that the widths of the slots produced are less than the diameter of the tool (1090 µm) in use. A marked decrease in the slot widths is particularly observed in Slots A and B, which are characterized by high burr density. Although this particular study does not directly encompass the concept of tool wear, the patterns that have emerged in the data give us grounds to predict its potential occurrence. Tool wear refers to the gradual loss or deformation of a tool’s innovation as a result of ongoing contact with the workpiece and subsequent material removal. More specifically, the machining parameters that have been applied in the context of Slots A and B seem to point towards the possibility of tool wear. The evidence lies in the fact that smaller slot widths are being produced, indicating that the tool might be wearing down and hence is unable to maintain the desired performance levels. The indicators in Slots A and B suggest that we might have to anticipate and accommodate tool wear, even though the current study does not actively factor it into its scope. This finding is crucial as it can potentially lead to the refinement of machining strategies for improved precision and also the timely maintenance or replacement of tools to ensure operational efficiency. Bhushan et al. [57] reported that an increase in the feed rate corresponded with amplified wear of the cutting tool. In other words, as the feed rate surged, there was a concurrent escalation in the deterioration of the cutting tool.

## 4. Conclusions

This study investigates the impact of machining parameters—cutting speed, feed rate, and depth of cut—on burr formation and slot width during micro-milling of AISI 316L material. The results indicate that decreasing feed rate and depth of cut reduce burr formation, in which increased burrs are associated with down-milling, and that feed rate is directly linked to burr formation; additionally, the study proposes an effective algorithm for quick and accurate detection of slot and burr widths, with potential implications for minimizing burr formation and enhancing tool maintenance strategies.

➢Burr formation during micro-milling of AISI 316L is not significantly affected by the cutting speed when the feed rate and depth of cut are kept constant.➢Decreasing the feed rate and depth of cut while keeping the cutting speed constant leads to a significant reduction in burr formation on both up-milling and down-milling sides due to a decrease in material deformation and displacement.➢The burr width on the down-milling side tends to be larger than on the up-milling side due to the continuous chip formation and compressive forces experienced during down-milling.➢Increasing the feed rate while holding other parameters constant notably enhances burr formation, suggesting the need for precise control of the feed rate for optimal machining outcomes.➢Larger burrs are more likely to occur on the down-milling side, as supported by the measurement of maximum burr widths, thus reinforcing the described burr formation mechanism.➢The proposed algorithm in the study can quickly and accurately detect both slot and burr widths for the up- and down-milling sides, surpassing potential human observational errors.➢The slot widths smaller than the tool diameter used indicate potential tool wear. This, along with the high burr density, suggests a decrease in tool performance, hinting at the need for improved machining strategies and tool maintenance.➢The study’s proposed approach could potentially minimize burr formation by providing a fast and accurate burr formation characterization that could be adapted to a closed-loop system. Future research could focus on refining this method and studying other factors that influence burr formation and tool wear.

## Figures and Tables

**Figure 1 micromachines-14-01376-f001:**
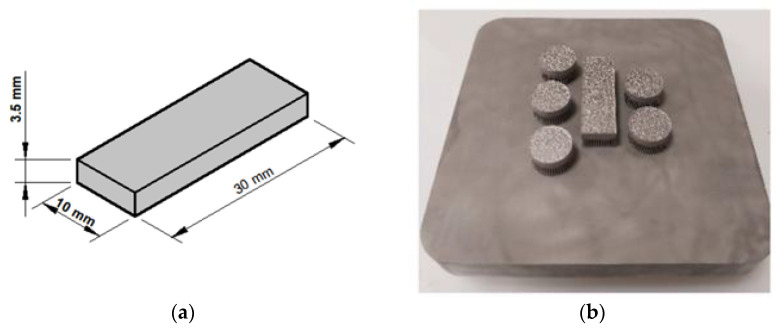
The procedure specimens: (**a**) dimensions of the specimens (mm), (**b**) manufactured AISI 316L specimens.

**Figure 2 micromachines-14-01376-f002:**
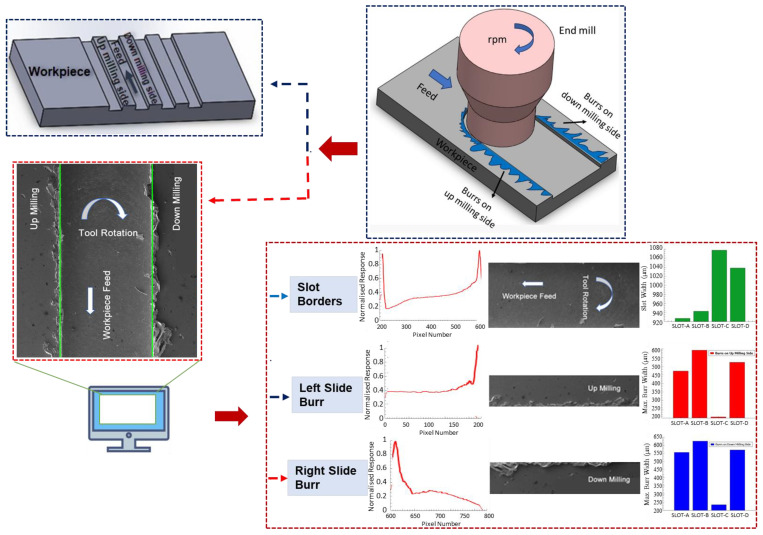
The Schematic view of the slot and burr measuring software shows the image input and measurement outputs.

**Figure 3 micromachines-14-01376-f003:**
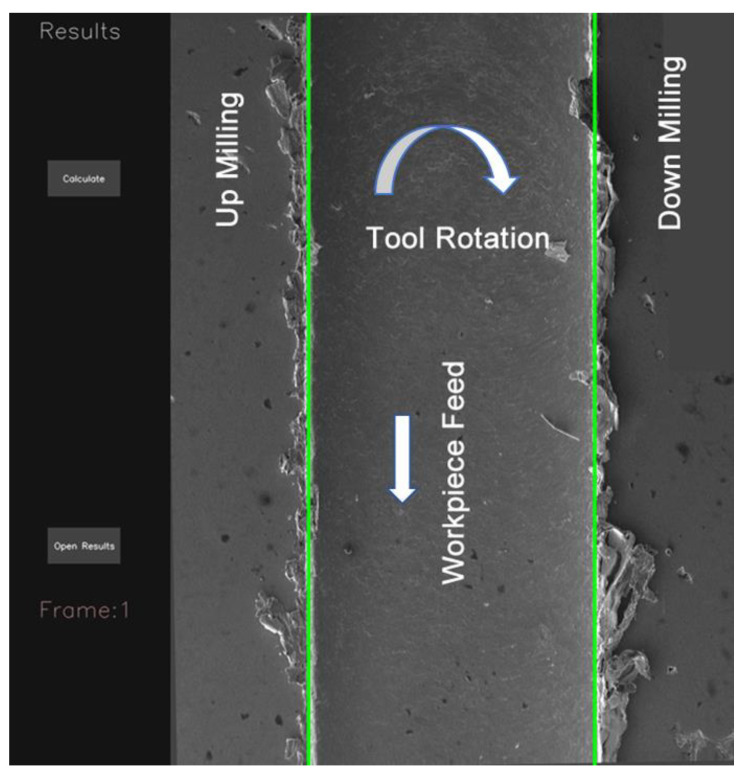
The Graphical User Interface (GUI).

**Figure 4 micromachines-14-01376-f004:**
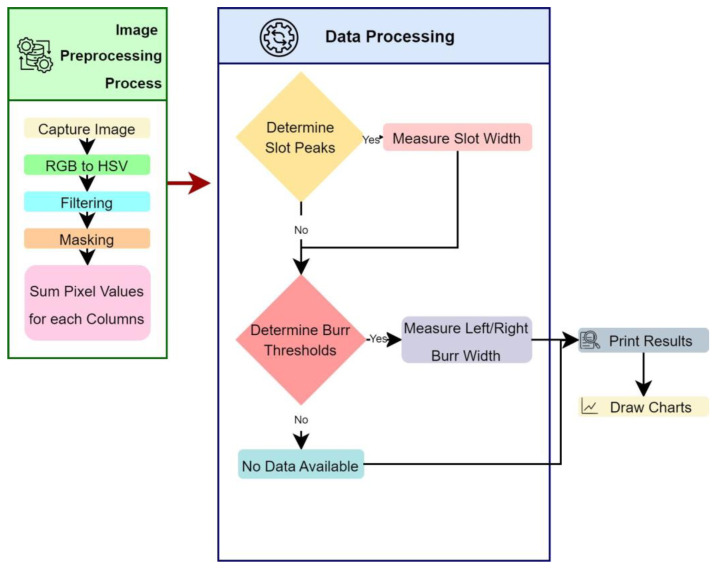
The flowchart of slot and burr characterization.

**Figure 5 micromachines-14-01376-f005:**
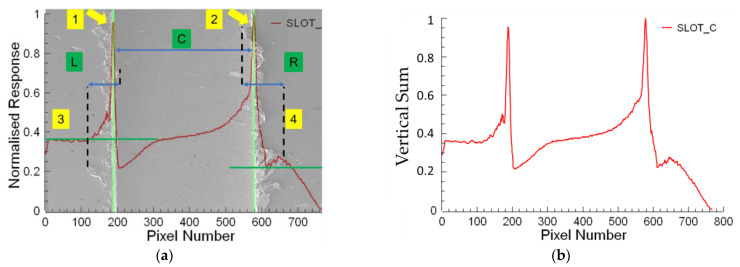
(**a**) Burr formations and slot length determined concerning the horizontal distance for the burr of the up-milling side, slot width, and burr of the down-milling side indicated with numbers 3-1, 1-2, 2-4, respectively; (**b**) processed image outputs for determining slot and burr formations.

**Figure 6 micromachines-14-01376-f006:**
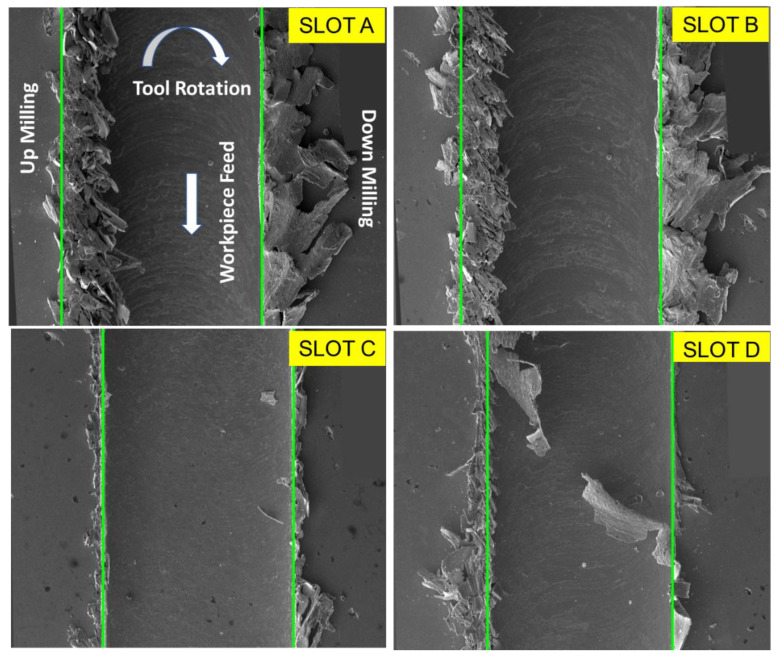
SEM images of burr formations and slots after micro-milling of AISI 316L material in different parameters.

**Figure 7 micromachines-14-01376-f007:**
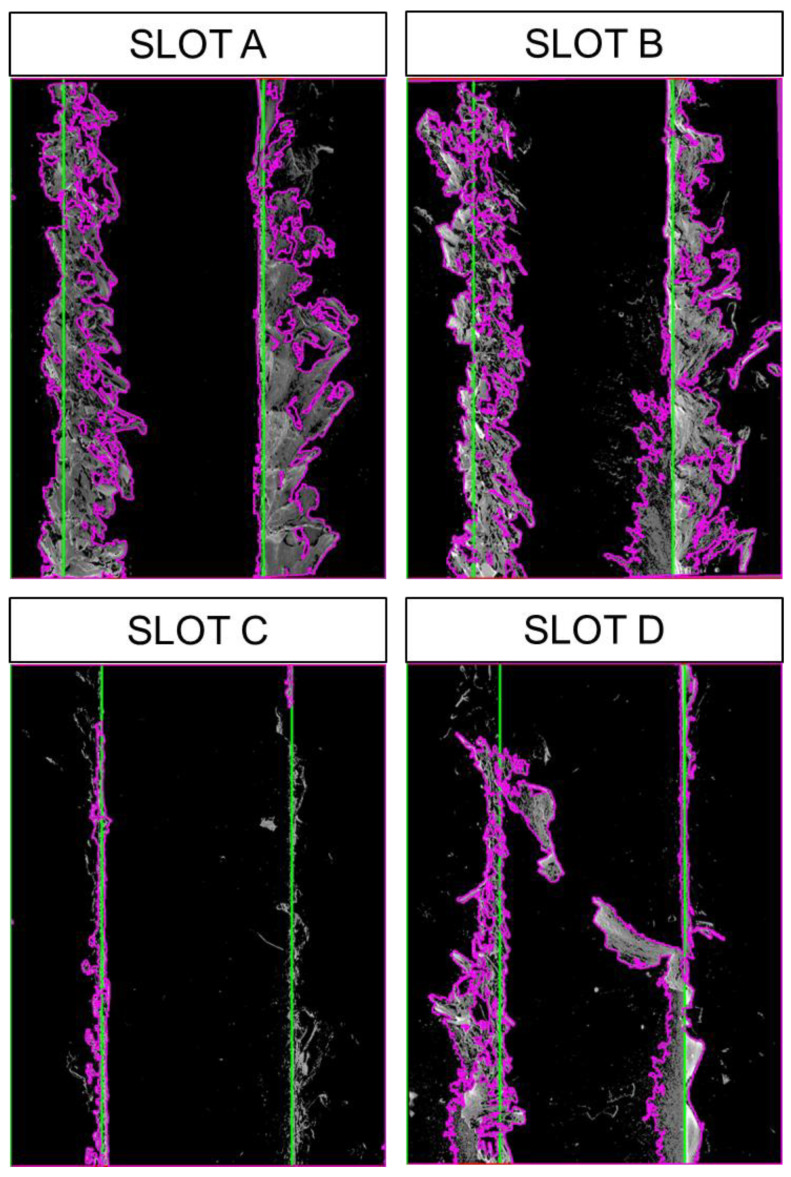
The determined slot widths and contoured burr formations.

**Figure 8 micromachines-14-01376-f008:**
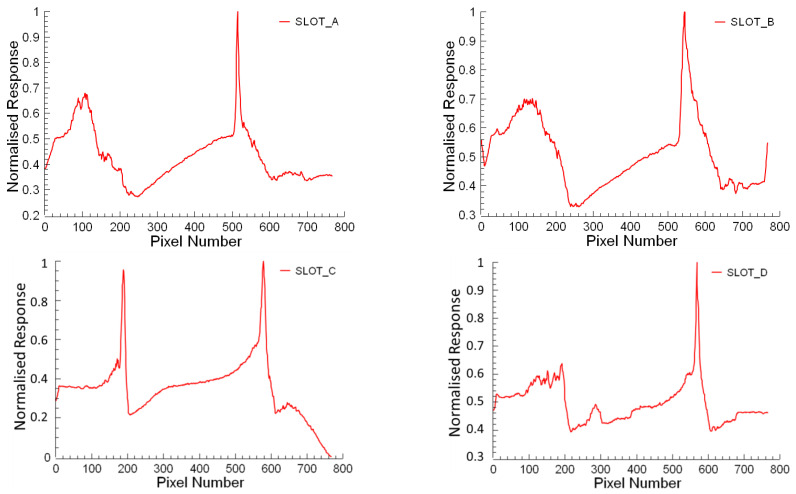
Processed image outputs for Slot A, B, C, and D of SEM images.

**Figure 9 micromachines-14-01376-f009:**
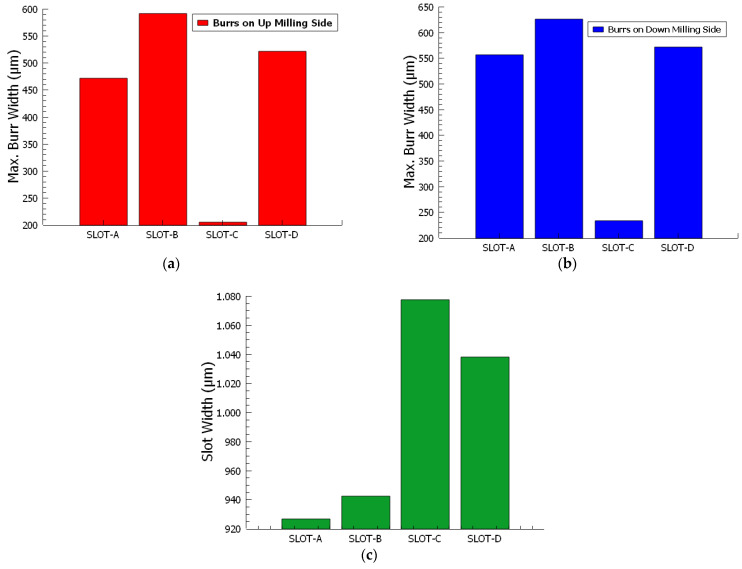
(**a**) The maximum burr widths formed on the up-milling sides. (**b**) The maximum burr widths formed on the down-milling sides. (**c**) Slot widths of SEM images.

**Table 1 micromachines-14-01376-t001:** The chemical composition of AISI 316L SS metal powder alloy [50].

Element	Cr	Ni	Mo	Mn
wt.%	16.5–18.5	10–13	2–2.25	0–2
**Element**	**Si**	**P**	**C**	**Fe**
wt.%	0–1	0–0.045	0–0.03	Remain

**Table 2 micromachines-14-01376-t002:** Applied parameters of the micro milling process.

SLOT-CODE	Cutting Speed (m/min)	Feed Rate (mm/min)	Depth of Cut (mm)
SLOT-A	15.7	25	0.5
SLOT-B	31.4	25	0.5
SLOT-C	31.4	10	0.2
SLOT-D	31.4	20	0.2

## Data Availability

Not applicable.

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
