# Peer review of "Image Processing Approach to Investigate the Correlation between Machining Parameters and Burr Formation in Micro-Milling Processes of Selective-Laser-Melted AISI 316L"

_micromachines, 2023, doi:10.3390/mi14071376_

Round 1
Reviewer 1 Report
Akkoyun et al. reported an image processing approach to investigate the correlation between machining parameters and burr formation in micro milling processes of selective laser melted AISI 316L. This work is interesting and can be published after a minor revision.
1. The authors used AISI 316L in this work. How about other stainless steel? Will observe similar fabrication principles?
2. The fronts in the Figures 2-3 should be larger.
3. Are Figures 3 and 6 the same in the inset of Figure 2?
4. It is better to add more descriptions on the interaction between laser and metal or additive manufacturing in the introduction section. The following references may be helpful. Prog. Electromagn. Res 2023, 176, 45; Frontiers in Chemistry 2021, 9, 823715; Exploration 2021, 1, 20210109.
Reviewer 2 Report
The article titled "Image Processing Approach to Investigate the Correlation Between Machining Parameters and Burr Formation in Micro-Milling Processes of Selective Laser Melted AISI 316L" provides valuable insights into the correlation between machining parameters and burr formation in the micro-milling of AISI 316L produced by selective laser melting. To enhance the manuscript, I propose the following suggestions:
Clearly express the study's objectives to direct readers and establish the research's context.
Consider incorporating statistical methods or instruments used to analyze the data and establish correlations between the machining parameters and burr formation.
Provide potential explanations or mechanisms for observed trends and correlations, based on scientific principles or previous research.
By incorporating these suggestions, the manuscript can be accepted.
